# Optimization of Efficient Development Modes of Offshore Heavy Oil and Development Planning of Potential Reserves in China

Taichao Wang [1,2], Fengming Liu [3] and Xin Li [4,*]

1   School of Energy Resources, China University of Geosciences, Beijing 100083, China
2   CNOOC Research Institute, Beijing 100028, China
3   GWDC, Panjin 124000, China
4   Research Institute of Petroleum Exploration and Development (RIPED), Beijing 100083, China
*   Correspondence: lixin1123@petrochina.com.cn

**Abstract:** Thermal recovery is still the most important means to increase heavy oil EOR. With the increase in the recovery factor and the difficulty of exploiting new exploration reserves, the efficient utilization of offshore heavy oil reserves has attracted much attention. However, due to the challenges of high development investments, high operating costs, platform safety factors, and high economic cumulative yield, the offshore heavy oil reserves of nearly 700 million tons have not been effectively utilized. In this paper, Chinese offshore heavy oil reserves were taken as the research object. The indoor one-dimensional experiments were carried out to optimize an applicable development method, and the superheated steam huff and puff was selected as the injection medium for high-speed and high-efficiency development of offshore heavy oil, which verified the great potential of the application of superheated steam in offshore heavy oil thermal recovery. A numerical simulation model for offshore heavy oil superheated steam injection development was established, and a dynamic model considering the thermal cracking of heavy oil was established through historical matching. Through the field numerical simulation models, the whole process development mode of a single sand body, thin interbedded reservoir superheated steam huff and puff turning to superheated steam flooding, and thick layer super heavy oil reservoir with bottom water sidetracking after superheated steam huff and puff for eight cycles was established. Through the numerical simulation method and grey correlation method, the main control factors of superheated steam development of different types of heavy oil reservoirs were determined, and the cumulative oil production charts of different types of reservoirs under the influence of the main control factors were built. The economic evaluation model of superheated steam development of offshore heavy oil was established. Combining multi-specialty of geological, reservoir engineering, drilling and completion, oceanographic engineering, economics, the economic limits of steam injection development under different reserve scales, and engineering conditions of offshore heavy oilfields were clarified. At last, we planned the economic production mode of undeveloped reserves and predicted the construction profile of superheated steam capacity of offshore heavy oil using the production charts and the economic charts. The research results clarify the great potential of thermal recovery development of offshore heavy oil, provide an important basis for the economic development of offshore heavy oil undeveloped reserves, and also provide an important decision for the sustainable and stable production of global heavy oil reservoirs.

**Keywords:** offshore heavy oil; superheated steam huff and puff; physical simulation experiment; numerical simulation; economic boundary; production capacity planning





## 1. Introduction

At present, the difficulty of heavy oil development in the world is gradually increasing [1–3]. Taking Chinese heavy oil development as an example, according to the statistics

at the end of 2021, the contribution of emerging technologies such as SAGD and in situ combustion had gradually increased, while the contribution of traditional steam stimulation had decreased, and the annual production capacity showed a decreasing trend at the beginning of the 21st century [4–6].

Due to the increasing domestic demand for energy and the increasing difficulty of heavy oil exploitation in the new exploration of onshore oilfields in China, how to efficiently utilize offshore heavy oil reserves has gradually attracted the attention of scholars from all over the world. China has extremely high heavy oil reserves in the sea. As of December 2022, the statistics showed that the heavy oil reserves with a viscosity greater than 350 mPa·s have reached nearly 700 million tons, of which more than 95% are located in the Bohai oilfield, which is located in the northeast sea area of China and belongs to the China National Offshore Oil Corporation [7,8]. It is the largest offshore oilfield and the largest heavy oil production base in China. By the end of 2022, the cumulative oil and gas equivalent of the Bohai oilfield had reached 500 million tons, but it was mainly light crude oil and natural gas, and the heavy oil production was only 20 million tons. Thermal recovery is considered to be the most mature method to improve the recovery factor of heavy oil. At present, only the Bohai NB35-2 oilfield and LD27-2 oilfield had carried out steam injection pilot tests for a few wells in China, relying on the developed platform, which had achieved good development results. There is no precedent for large-scale thermal recovery of offshore heavy oil [9–11]. Among them, the LD27-2 oilfield was put into production in 2011 and developed using two wells for steam stimulation. By the end of 2022, the cumulative oil production had exceeded 130 thousand tons. The NB35-2 oilfield was put into production in 2012 and developed using low-temperature multi-element thermal fluids. Some well groups had been converted to steam flooding in the later stage. By the end of 2022, the cumulative oil production had exceeded 450 thousand tons. From the development effect of the pilot test area, offshore thermal recovery of heavy oil has shown great potential.

Due to economic and technical constrains, the main challenges faced by large-scale thermal recovery of offshore heavy oil mainly include: (1) The large distance between the offshore platform and the target layer. For a 1000-m reservoir, the drilling footage exceeds 3000 m, which brings extremely high drilling and completion costs [12]. (2) The geological reservoir conditions of offshore heavy oil are poor; more than 85% of the Bohai heavy oil reserves are at a depth of 1000 m to 1500 m, and the reservoirs are generally in the middle or deep layers, with high original formation pressure. Directional wells or horizontal wells are used for development, with larger well bore depths and higher heat losses. In addition, from the perspective of the oil–water relationship, about 57% of the reserves are widely developed with edge and bottom water, and water channeling is much easier to occur in the process of production, resulting in greatly reduced thermal recovery effect [13]. (3) The development investment is high, and the economic cumulative production limit per well is high. Comparing the thermal recovery of offshore heavy oil with the thermal recovery of onshore oilfields, the development strategy of "less wells and higher production" must be adopted because of the small space of offshore platforms, the difficulty of offshore operations, the small number of platform wells, and the high operating costs. According to the data comparison, compared with the thermal recovery development of onshore oilfields, the design well spacing of offshore heavy oil is 2 to 3 times, the well control reserves are 5 to 10 times, the thermal recovery engineering investment is 13 times, the drilling and completion investment is 9 times, and the economic cumulative oil production per well is about 10 times of that of onshore oilfields. Therefore, it is very important for the large-scale development of offshore heavy oil reserves to optimize the economic development method suitable for offshore heavy oil thermal recovery and formulate efficient development strategies [14–17].

At present, there are few research results related to large-scale thermal recovery of offshore heavy oil in the industry, especially few related research foundations for the economic limit of thermal recovery of offshore heavy oil [18]. In this paper, the Bohai oilfield

in China was taken as the research object. Firstly, through the indoor one-dimensional physical simulation experiment, the thermal recovery injection medium was optimized. Through the numerical simulation method, the accurate characterization model of the thermal recovery numerical simulation of heavy oil was built. By using the actual field model, the reasonable development modes of different viscosity and different types of reservoirs were optimized. Combined with the numerical simulation method and the grey correlation method, the main control factors affecting the cumulative oil production of different types of reservoirs were screened and the influence charts of the main control factors were built. The combination of multiple disciplines formed the economic limit of thermal recovery of offshore heavy oil under different engineering modes, and the economic production mode of offshore heavy oil reservoirs was planned to predict the production profile of offshore heavy oil.

## 2. Experiment

### 2.1. Experimental Purpose

Through the indoor one-dimensional physical simulation experiment, the comparison of the recovery factor of crude oil huff and puff with different steam injection mediums and different crude oil viscosity were explored. The injection mediums included normal atmospheric temperature water (25 °C), hot water (120 °C), saturated hot water (250 °C), saturated steam (250 °C), and superheated steam (300 °C). Heavy oils with viscosities of 350 mPa·s (JZ23-2 heavy oil), 3000 mpa·s (LD21-2 heavy oil) ordinary heavy oil, and 50,000 mpa·s (LD 5-2N heavy oil) were selected as the research objects. The basic parameters of the sand filling model are shown in Table 1.

**Table 1.** Basic geological and reservoir parameters of different heavy oil and experiment models.

| Oil Sample | Type of Oil Sample | Oil Viscosity at 50 °C, mPa·s | Porosity, % | Permeability, mD | Oil Saturation, % | Initial Pressure, MPa |
|---|---|---|---|---|---|---|
| JZ23-2 | Common heavy oil | 350 | 29.0 | 668 | 68.1 | 11.1 |
| LD21-2 | Common heavy oil | 2980 | 32.1 | 2480 | 68.0 | 15.1 |
| LD5-2N | Super heavy oil | 50,154 | 35.0 | 2894 | 75.2 | 12.0 |

### 2.2. Experiment Apparatus

The setup of the one-dimensional steam huff and puff experiment is shown in Figure 1. The huff and puff experiment was performed in a one-dimensional sandpack model (3.8 cm in diameter and 48 cm in length) made of Hastelloy C276. Its maximum operating pressure and temperature were 50 MPa and 450 °C, respectively. A pump was used to inject water and heavy oil into the sandpack model. Its minimum injection rate can reach 0.01 mL/min. A steam generator was used to heat the water provided by the pump to the specified injection temperature, and its maximum working pressure and the temperature reached were 35 MPa and 450 °C, respectively. A temperature control system was used to control the sandpack model temperature by six belt heaters (power 3500 W) surrounded by the sandpack surface, and a pressure probe together with a computer was used to monitor the sandpack pressure. The products were cooled to 60 °C through the cooling system and condenser, to avoid damage to the back-pressure regular (BPR). The BPR together with a hand pump was used to adjust the sandpack model pressure. The gas–liquid separator separated the products into the liquid phase and gas phase. An electronic balance and a gas meter were used to measure the mass of the liquid and the gas volume.

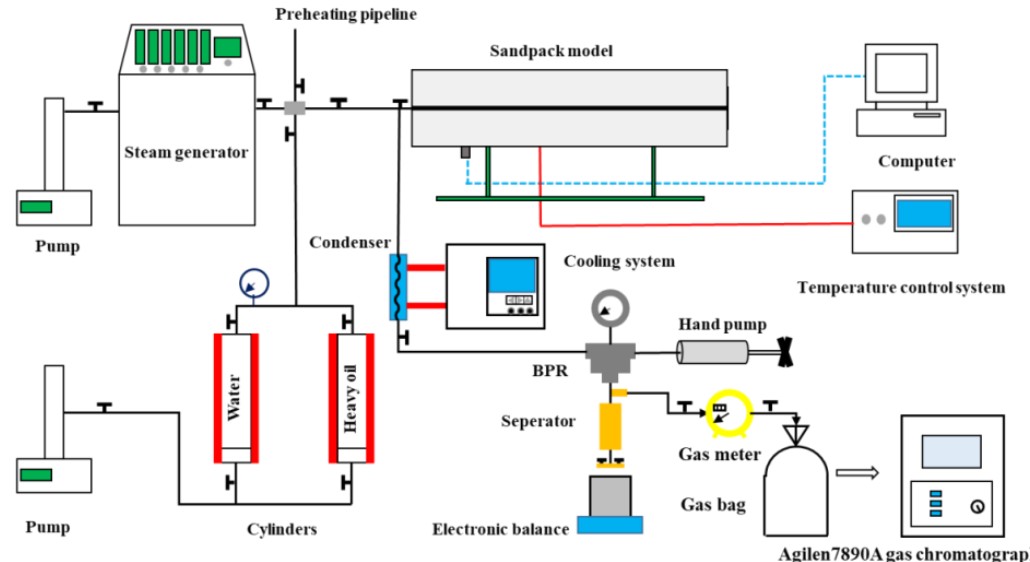

**Figure 1.** Schematic illustration of the experimental setup for one-dimensional huff and puff experiment.

*2.3. Experiment Procedures*

First, in the small-scale core model, the sand samples were proportioned according to the porosity and permeability of the target experiment. Then, the mixture sand sample was filled into the clean one-dimensional sandpack model. After the experimental setups were connected, an inspection for leakage and vacuuming were completed. The temperature control system controlled the band heater to heat the sandpack model to the reservoir temperature. Then, the pump was used to inject water and heavy oil into the sandpack model, and the initial oil saturation was measured. After the sandpack model preparation was finished, the steam heated the distilled water to the predetermined temperature, and the injection medium was injected into the model at a rate of 10 mL/min until the reservoir pressure reached the desired pressure. After medium injection, the model was soaked until the pressure was stable. The time of medium injection and soaking in each cycle was 1.5 to 2 h in total. During production, the BPR was used to control the outlet pressure, and 0.5 to 1 MPa/h was applied as a pressure drop rate in each cycle. The whole cycle finished when the model pressure was 1 MPa. The huff and puff cycles were repeated until the end of the eighth cycle. The mass of liquid and gas volume was measured every 10 min. After the experiment, the core was extracted to measure the residual oil saturation.

*2.4. Experimental Results and Discussion*

Tables 2–4 shows the result of one-dimensional huff and puff experiment. It can be seen from table that for heavy oil with different viscosities, the higher the viscosity, the lower the recovery efficiency. The water flooding effect at normal temperatures was poor, with a recovery factor of only 3.3% to 21.2%. Under different viscosities, the recovery rate was still low and the residual oil saturation was high; when saturated steam or superheated steam was injected, the recovery factor was significantly improved, and the latent heat of vaporization of the steam released more heat than that of hot water. At the same time, comparing the effects of superheated steam and saturated steam under different viscosities of heavy oil, the recovery factor of heavy oil with viscosities of 350 mPa·s, 2980 mPa·s, and 50,154 mPa·s increased by 11.2%, 23.3%, and 39.1%, respectively. The higher the viscosity, the better the development effect of superheated steam and the greater the decrease in residual oil saturation, which verified the great potential of superheated steam in thermal recovery of offshore heavy oil.

**Table 2.** Experimental results of one-dimensional huff and puff of oil with a viscosity of 350 mPa·s under different injection media.

| Injection Medium | Experimental Method | Residual Oil Saturation, % | Recovery Factor, % |
|---|---|---|---|
| atmospheric temperature water (25 °C) | | 33.2 | 21.2 |
| hot water (120 °C) | | 23.0 | 33.2 |
| saturated hot water (250 °C) | Huff and puff for 8 cycles | 14.4 | 43.4 |
| saturated steam (250 °C) | | 8.4 | 52.6 |
| superheated steam (300 °C) | | 6.8 | 58.5 |

**Table 3.** Experimental results of one-dimensional huff and puff of oil with a viscosity of 2980 mPa·s under different injection media.

| Injection Medium | Experimental Method | Residual Oil Saturation, % | Recovery Factor, % |
|---|---|---|---|
| atmospheric temperature water (25 °C) | | 39.3 | 18.5 |
| hot water (120 °C) | | 25.4 | 27.3 |
| saturated hot water (250 °C) | Huff and puff for 8 cycles | 16.4 | 38.4 |
| saturated steam (250 °C) | | 11.5 | 42.0 |
| superheated steam (300 °C) | | 7.1 | 51.8 |

**Table 4.** Experimental results of one-dimensional huff and puff of oil with a viscosity of 50,154 mPa·s under different injection media.

| Injection Medium | Experimental Method | Residual Oil Saturation, % | Recovery Factor, % |
|---|---|---|---|
| atmospheric temperature water (25 °C) | | 57.5 | 3.3 |
| hot water (120 °C) | | 46.5 | 9.9 |
| saturated hot water (250 °C) | Huff and puff for 8 cycles | 27.9 | 16.9 |
| saturated steam (250 °C) | | 19.9 | 21.5 |
| superheated steam (300 °C) | | 13.8 | 29.9 |

## 3. Numerical Simulation

### 3.1. Matching of the Experimental Data

3.1.1. Building the One-Dimensional Numerical Simulation Model

In this study, CMG-STARS software was used to develop a numerical model to simulate the one-dimensional superheated steam injection experiment. According to the size of the one-dimensional sandpack model, a one-dimensional numerical simulation model was established as shown in Figure 2. A reasonable grid size and step size can not only approach reality, but also improve the operational speed of the grid model. Therefore, the one-dimensional numerical simulation model included 48 × 19 × 19 (17,328) grids, and the grid size was 1 cm (X) × 0.2 cm (Y) × 0.2 cm (Z). The reservoir properties (porosity, permeability, oil saturation) were obtained from the experimental data.

The fluid model was set up with five components: water, heavy oil, light oil, $H_2S$, and $CH_4$. The kinetic conversion relationship used in the numerical simulation is shown in Equations (1) and (2) to characterize the process of water thermal cracking during the steam injection [5,19–22].

$$\text{Heavy oil} \xrightarrow{\text{Spiltting}} \text{Light oil} + CH_4 \tag{1}$$

$$\text{Heavy oil} + \text{Steam} \xrightarrow{\text{Aquathermolysis}} \text{Light oil} + CO_2 \tag{2}$$

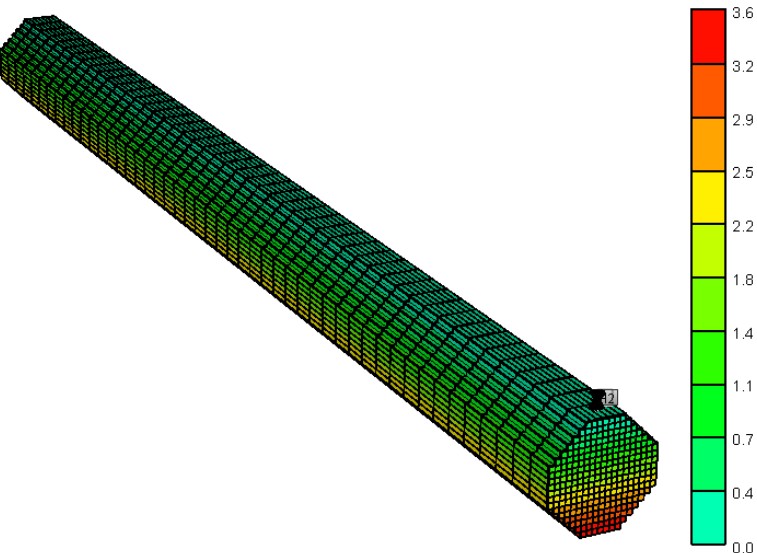

**Figure 2.** One-dimensional numerical simulation model.

### 3.1.2. History Matching

Taking the cumulative oil production and water production of the one-dimensional superheated steam experiment as the matching targets, the matching was carried out by mainly adjusting the relative permeability curve and kinetic parameters, such as activation energy and frequency factor in the developed numerical simulation model. As shown in Figures 3–5, the simulated cumulative oil production and water production were consistent with those obtained by experiments. The average absolute errors between the measured and simulated cumulative oil production were 1.79%, 6.58%, and 2.37%, respectively, and those of cumulative water production were 4.55%, 2.03%, and 2.90%. All the results indicated that the measured and simulated liquid production had good consistency and the developed model can accurately predict oil production during huff and puff processes. Therefore, the calibrated parameters were used to predict oil production in the following reservoir numerical simulations.

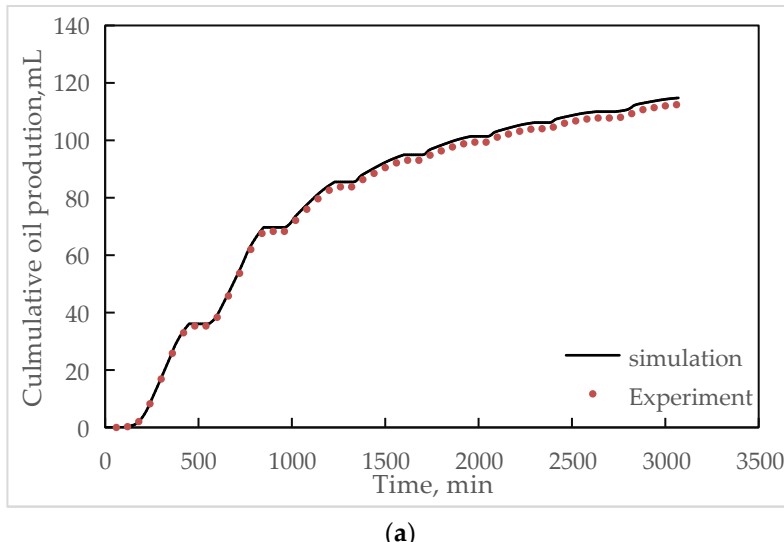

(**a**)

**Figure 3.** *Cont*.

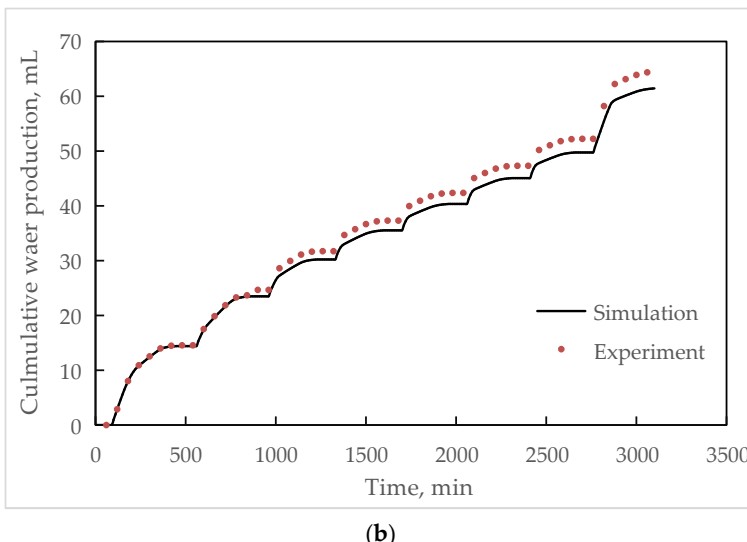

(**b**)

**Figure 3.** The matching of measured and simulated results of JZ23-2 heavy oil: (**a**) cumulative oil production; (**b**) cumulative water production.

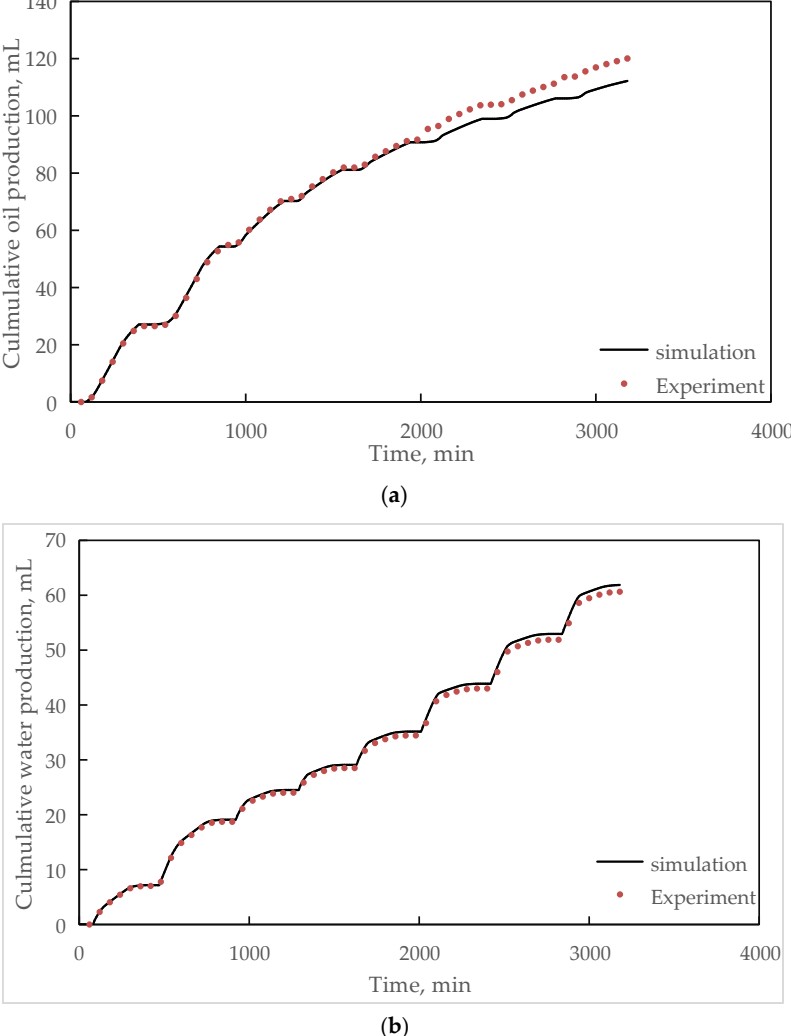

**Figure 4.** The matching of measured and simulated results of LD21-2 heavy oil: (**a**) cumulative oil production; (**b**) cumulative water production.

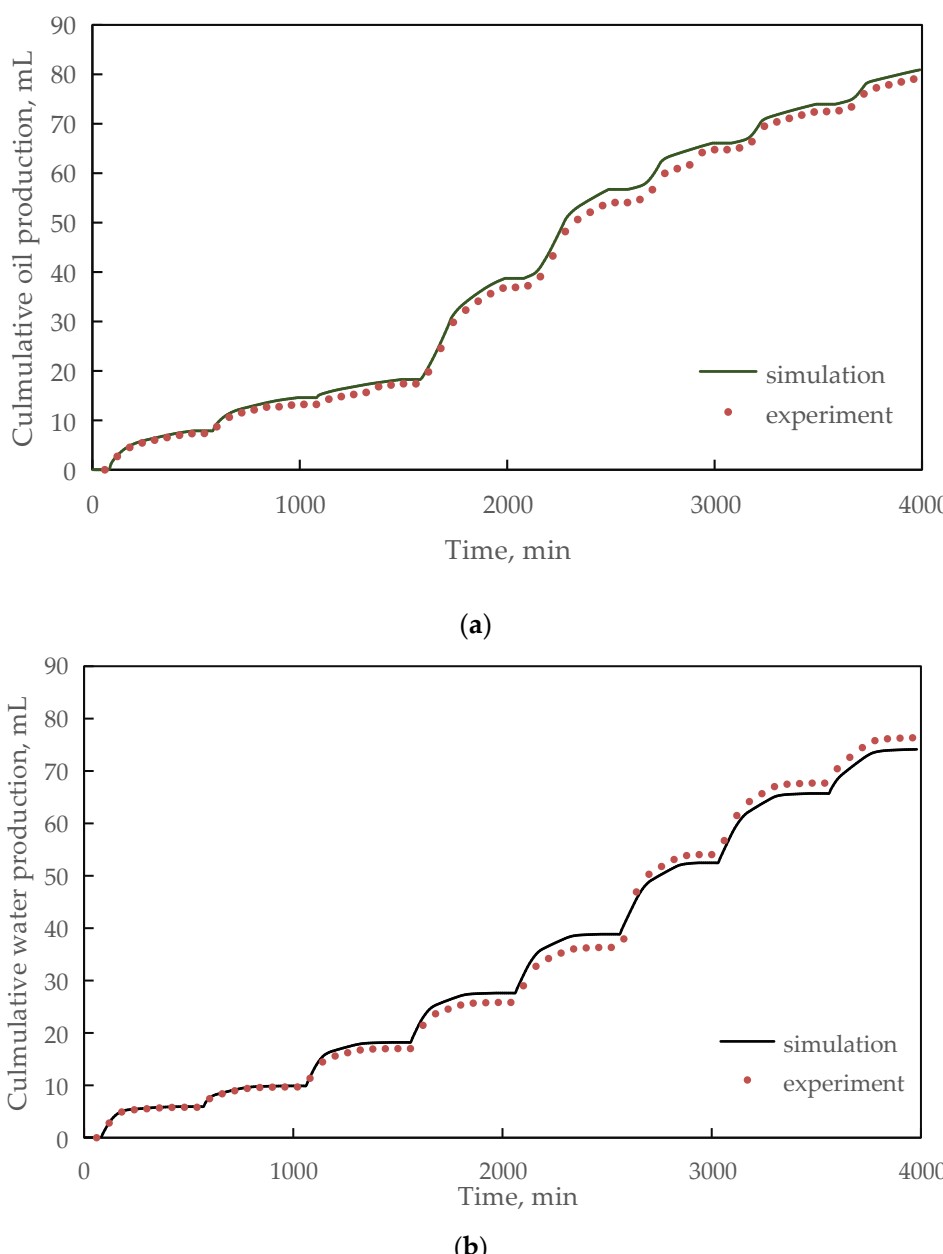

**Figure 5.** The matching of measured and simulated results of LD5-2N super heavy oil: (**a**) cumulative oil production; (**b**) cumulative water production.

*3.2. Development Mode Optimization*

The three-dimensional reservoir scale models were established based on the actual reservoir and fluid parameters of the target oilfield. The basic parameters of the reservoir model are shown in Table 5. In addition, the relative permeability curve and the kinetic parameters of aquathermolysis, which were well fitted in the one-dimensional numerical simulation, were also applied to the three-dimensional model. According to the one-dimensional simulation experiment, the superheated steam was used as the injection medium, and the superheated steam huff and puff was used as the initial development mode. Combined with the three-dimensional reservoir numerical simulation, the different replacement development mode of the different reservoir types in the later stage were optimized.

**Table 5.** The key parameters of the reservoir scale model and comparison schemes.

| Oilfield | JZ23-2 | LD21-2 | LD5-2N |
|---|---|---|---|
| Grid size (X·Y·Z), m | 20 × 23 × 1 | 15 × 18 × 1 | 20 × 21 × 1 |
| Number of grid (X·Y·Z), number | 112 × 253 × 48 | 162 × 42 × 52 | 202 × 139 × 55 |
| Designed number of wells, number | 56 | 16 | 28 |
| Reservoir types | Layered reservoir | Single sand body reservoir | Thick layer of super heavy oil reservoir |
| Well types | Directional well | Horizontal well | Horizontal well |
| Reservoir thickness, m | 38 | 20 | 40 |
| Depth of burial, m | 998 | 1396 | 897 |
| Reserve volume, $10^4$ m$^3$ | 2824 | 1025 | 2801 |
| Designed well spacing, m | 200~220 | 180~200 | 125~150 |
| Comparison scheme | Superheated steam huff and puff for 16 cycles (1), switching to superheated steam flooding after superheated steam huff and puff for 4 cycles (2), switching to superheated steam flooding after superheated steam huff and puff for 8 cycles (3), sidetracking after superheated steam huff and puff for 8 cycles (4). Corresponding to the coordinate axis in the following figures. | | |

### 3.2.1. Single Sand Body Reservoir

Figure 6 shows the comparison results of the recovery factor of single sand body reservoir (LD 21-2) under different development modes; it can be seen that the recovery factor of switching to superheated steam flooding after superheated steam huff and puff for 8 cycles was the highest, followed by superheated steam flooding after 4 cycles, and the recovery factor of huff and puff for 16 cycles was the lowest. This is because a large amount of remaining oil between injection and production wells can be driven out after switching to steam flooding, which is difficult to achieve by huff and puff [23,24]. Therefore, for a single sand reservoir, it is recommended to adopt the whole process development mode of switching to superheated steam flooding after eight cycles of superheated steam huff and puff.

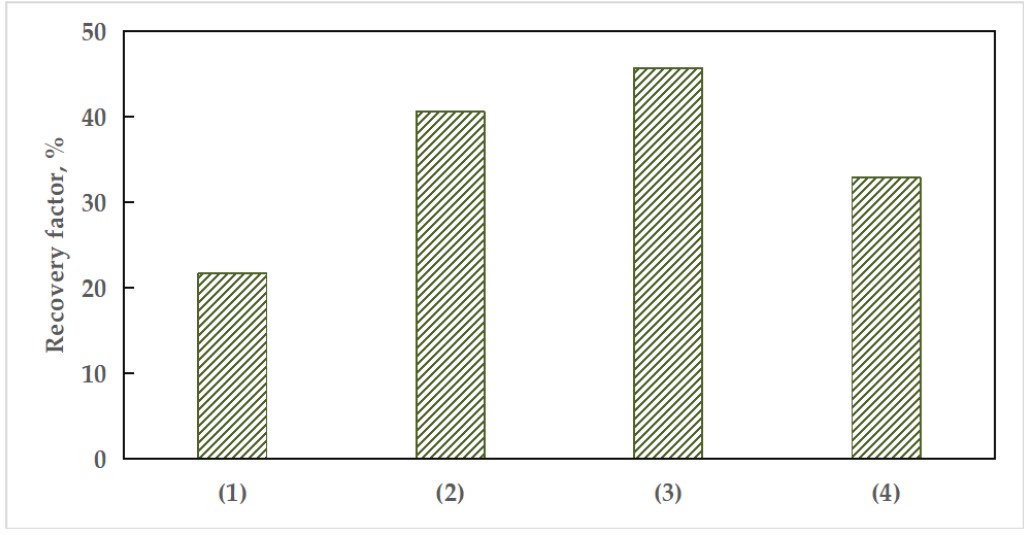

**Figure 6.** Comparison results of recovery factor of single sand body reservoir.

### 3.2.2. Layered Reservoir

Figure 7 shows the comparison results of recovery factor of layered reservoir under different development modes. It can be seen that for layered reservoirs, the trend of the recovery factor was basically the same as that of single sand body reservoir. Therefore, for common heavy oil in offshore oilfields, superheated steam flooding is also the best method to improve the development effect after high-cycle steam huff and puff.

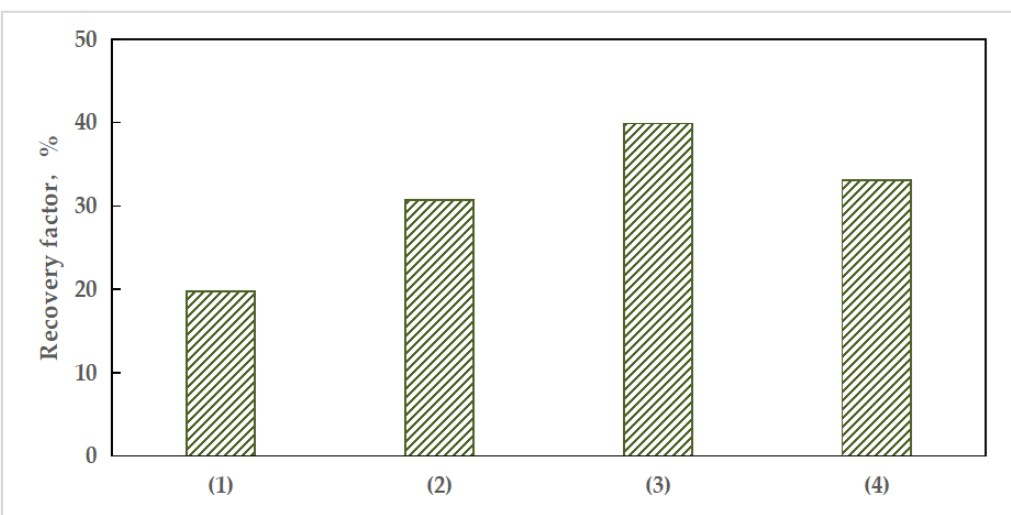

**Figure 7.** Comparison results of recovery factor of layered reservoir.

### 3.2.3. Thick Layer of Super Heavy Oil Reservoir

Figure 8 shows the comparison results of the recovery factor under different development modes of the thick layer of a super heavy oil reservoir. It can be seen that the development effect of switching to sidetracking huff and puff was better than switching to superheated steam flooding. This is because the super heavy oil steam flooding has a high actuating pressure gradient under the condition of large offshore well spacing. A $\varphi$ 2.54 cm $\times$ 50 cm one-dimensional sand packing pipe model was used to measure the actuating pressure gradient changing of LD5-2N super heavy oil at different temperatures, as shown in Figure 9. It can be seen from the figure that when the temperature reached 80 °C, the pressure gradient increased abruptly and remained unchanged when reaching the maximum value, indicating that the fluid did not flow and could not be displaced. When the temperature reached 100 °C, the pressure gradient rose slowly and remained unchanged when reaching the maximum value, which indicates that the fluid had viscoelastic deformation and could be displaced but the flow was still not smooth. When the temperature reached 120 °C to 150 °C, the pressure gradient first increased and then decreased slightly, indicating that the fluid could flow in this temperature range and realize effective displacement [25,26].

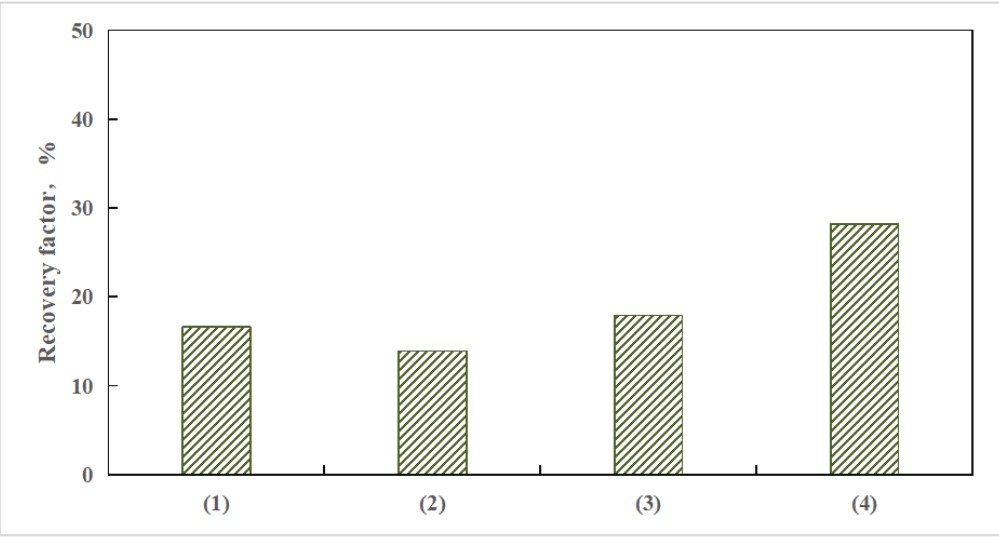

**Figure 8.** Comparison results of recovery factor of LD5-2N super heavy oil.

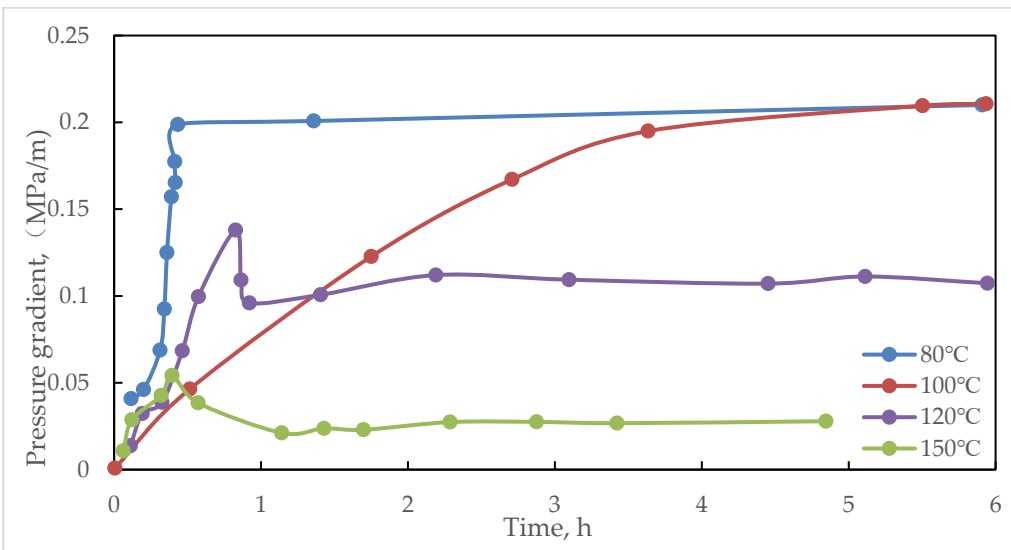

**Figure 9.** The pressure gradient of LD5-2N super heavy oil single pipe displacement experiment at different temperatures.

The small-scale numerical simulation technology was used to simulate the displacement process, so as to determine the start-up temperature of steam flooding under different conditions. The principle of the numerical simulation of a small-scale model was based on a group of completed single pipe displacement experiments; a numerical simulation model completely consistent with the experiment size was established, and the physical process of the displacement experiment was matched. When the simulated pressure gradient coincided with the experimental pressure gradient, the prediction process of small-scale numerical simulation represented the experimental process, and the corresponding temperature when the pressure gradient slowly rose was the start-up temperature. On this basis, the steam flooding numerical simulation of LD5-2N super heavy oil models under different well spacings were designed. As shown in Figure 10, it can be seen that the start-up temperature increased continuously with increasing well spacing.

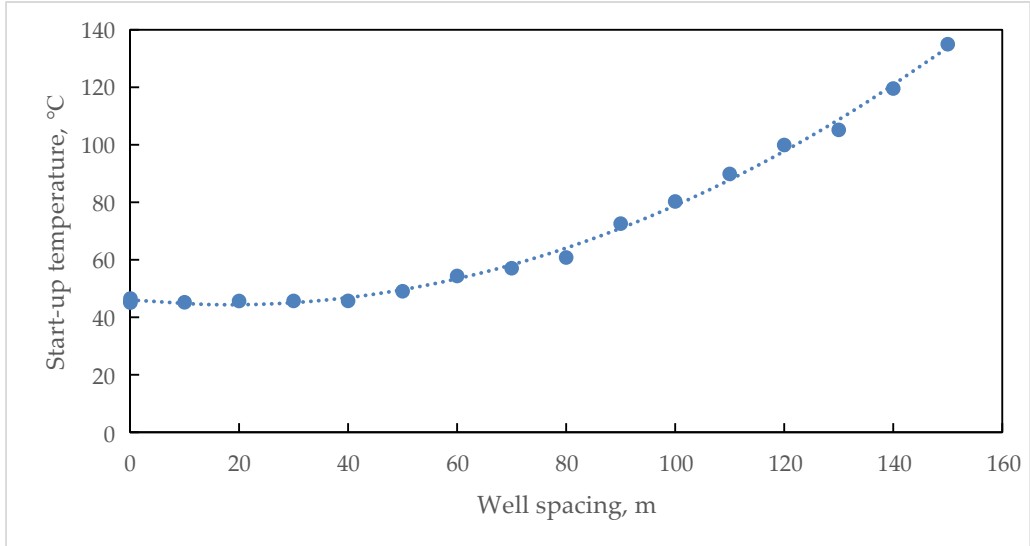

**Figure 10.** Numerical simulation result of changing start-up temperature of LD5-2N super heavy oil with different well spacings.

The prediction model of the bottom hole temperature of LD5-2N super heavy oil after different huff and puff cycles was established, as shown in Figure 11. With the increasing number of cycles, the bottom hole temperature gradually increased. However, the temperature after the eighth cycle was 96 °C, which was still lower than the start-up temperature (105 °C~134 °C) under the designed well spacing (120 m~150 m). Therefore, the technical limitations of steam flooding after steam huff and puff of offshore super heavy oil can be explained.

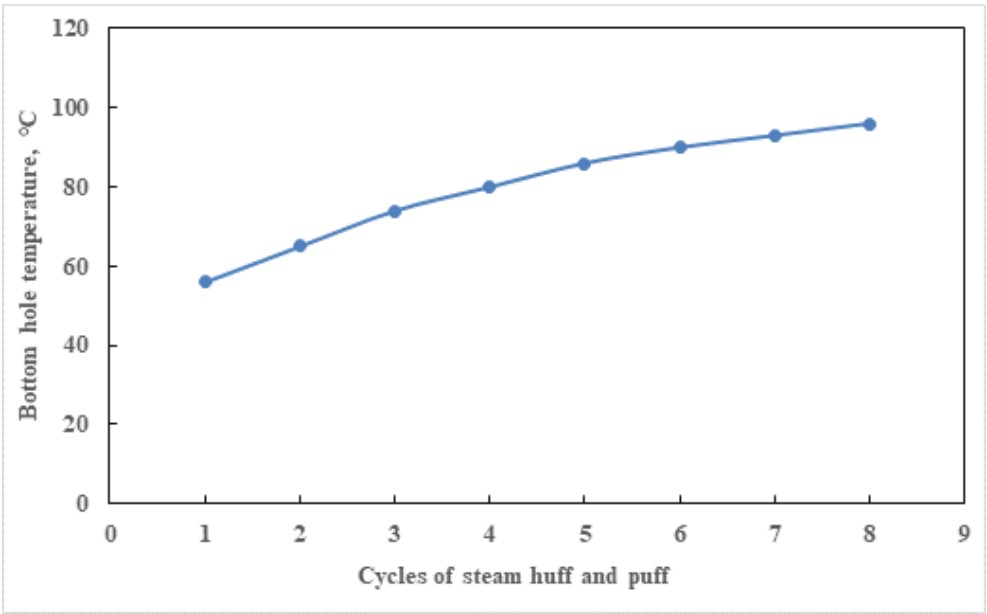

**Figure 11.** The change in bottom hole temperature of super heavy oil after different numbers of cycles huff and puff cycles.

### 3.3. Sensitivity Analysis

Combined with the classification of undeveloped reserves, the statistics of key geological reservoir parameters affecting steam injection development are shown in Table 6. Two methods were adopted for the analysis of the main control factors: the first method was to use the CMOST module in the reservoir numerical simulation software CMG to conduct sensitivity analysis and establish the influence chart of the main control factors. The analysis results of the main control factors are shown in Figure 12. It can be seen from Figure 12 that the main controlling factors of single sand body reservoirs are reservoir thickness and oil viscosity. The main controlling factors of layered reservoirs are the net-to-gross ratio and reservoir thickness. The main controlling factors of the thick layer of extra and super heavy oil reservoirs are reservoir thickness and water energy.

**Table 6.** Key geological reservoir parameters of heavy oil in different reservoir types.

| Reservoir Types | Reference Pressure, MPa | Reservoir Thickness, m | Oil Saturation | Oil Viscosity, mPa·s | Permeability, mD | Water Energy (times) | Net-to-Gross Ratio (NTG) |
|---|---|---|---|---|---|---|---|
| Single sand body reservoir | 6.5~16.0 | 4~40 | 0.5~0.68 | 350~3000 | 300~5000 | 0.1~7 | |
| Layered reservoir | 6.5~16.0 | 10~50 | 0.5~0.68 | 350~5000 | 300~3000 | 0.1~10 | 0.3~0.9 |
| Thick layer of extra and super heavy oil reservoir | 6.5~14.0 | 20~60 | 0.5~0.9 | 10,000~50,000 | 2000~5000 | 10~50 | |

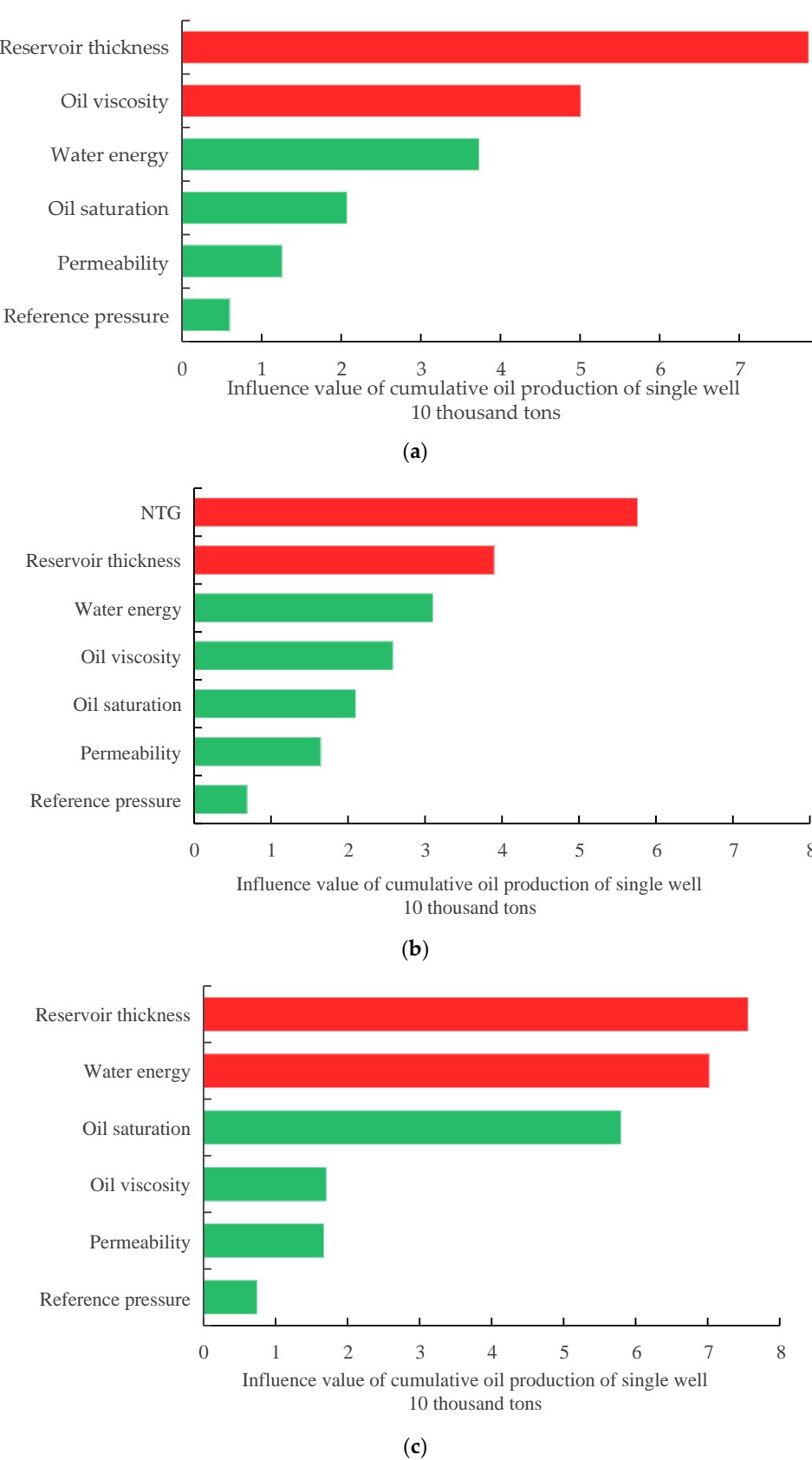

**Figure 12.** Influence chart of main control factors for steam injection development of different types of reservoirs. (**a**) Influence chart of sensitivity analysis of single sand body reservoir. (**b**) Influence chart of sensitivity in Thin layered reservoir. (**c**) Influence chart of sensitivity in thick layer of extra and super heavy oil.

The other method is to use the grey correlation analysis method, which was proposed in the 1980s to study the correlation degree of various factors in the dynamic development of the system [27–29]. This method does not require the data to obey some typical probability distribution law, which is simple and practical. With continuous development and progress, this method is also used to compare the similarity of any two sequences or two curves, so it is widely used in scientific research. In the application of the grey correlation analysis method, in order to eliminate the influence of dimensional differences between data on the results, a dimensionless analysis is required. The commonly used methods are the initial value method, equalization method, maximum method, minimal method, and interval analysis method. However, different processing methods have a great impact on the results of the grey correlation analysis method, and may produce conflicting results. In recent years, the slope correlation method proposed from the perspective of similarity definition overcame this shortcoming. The relative size of the calculation results will not differ due to different dimensionless processing methods, which greatly increases the reliability of the calculation results. The basic principle is to refer to and compare the differences of the relative change rate of the sequence, and to compare the correlation degree between the reaction sequences.

The calculation method of the slope correlation algorithm is:

$$\varepsilon_i(k) = \frac{1}{1 + \left| \frac{x_0(k)}{x_0(k)} - \frac{\Delta x_i(k)}{x_i(k)} \right|}$$

$$r_{(x_0, x_i)} = \frac{1}{N-1} \sum_{K=1}^{N-1} \varepsilon_i(k)$$

Additionally, $\Delta x_i(k) = x_i(k+1) - x_i(k)$, $i = 0, 1, \ldots, M$; $k = 1, 2, \ldots, N - 1$, where $\varepsilon_i$ is the correlation number; $x_0$ is the reference sequence; $x_i$ is a comparison sequence; M is the number of comparison virtual columns; $i$ and $k$ are the calculation serial numbers; N is the number of data points in the sequence; and $r_{(x_0, x_i)}$ is the slope correlation between the sequences $x_0$ and $x_i$.

The magnitude of the slope correlation degree value reflects the magnitude of the correlation degree between the sequences. The larger the value of the slope correlation degree, the greater the correlation degree between the two sequences. The larger value of the slope correlation degree, the greater the correlation between the two sequences. In the analysis of the main control factors, the cumulative oil production was used as the reference sequence, and the key parameters of each geological reservoir were the comparison sequence. The greater the correlation between the comparison sequence and the reference sequence, the greater the influence of this factor on the cumulative oil production. Table 7 shows the basic data of the different types of reservoirs, Table 8 is the dimensionless processing of each basic data using the mean method, and Table 9 is the correlation degree value calculated by the slope correlation degree method.

It can be seen from the calculation results that the main control factors of the different types of reservoirs are different: the main control factors of single sand body reservoir are reservoir thickness and oil viscosity; the main control factors of layered reservoirs are reservoir thickness and net-to-gross ratio; and the main control factors of the thick layer of extra and super heavy oil reservoirs are reservoir thickness and water energy. From the results of the two sensitivity analysis methods, the sensitivity factors of the different types of reservoirs corresponding to different analysis methods are basically the same, so the main control factors selected were used as the basis for the following research task.

**Table 7.** Geological parameters of different types of oil reservoirs.

| Reservoir Types | Reservoir Pressure MPa | Reservoir Thickness m | Oil Saturation | Oil Viscosity mPa·s | Permeability mD | Water Energy | NTG | Cumulative Oil Production 10 Thousand Tons |
|---|---|---|---|---|---|---|---|---|
| Single sand body reservoir | 6.5 | 4 | 0.50 | 350 | 300 | 0.1 | | 7.7 |
| | 8 | 8 | 0.54 | 500 | 1000 | 1.0 | | 10.2 |
| | 10 | 10 | 0.57 | 750 | 2000 | 2.0 | | 12.9 |
| | 12 | 20 | 0.61 | 1000 | 3000 | 3.0 | | 16.0 |
| | 14 | 30 | 0.64 | 2000 | 4000 | 5.0 | | 17.2 |
| | 16 | 40 | 0.68 | 3000 | 5000 | 7.0 | | 19.5 |
| Layered reservoir | 6.5 | 4 | 0.50 | 350 | 300 | 0.1 | 0.3 | 5.0 |
| | 8 | 8 | 0.54 | 750 | 1000 | 1.0 | 0.5 | 5.2 |
| | 10 | 10 | 0.57 | 1000 | 1500 | 2.0 | 0.6 | 6.0 |
| | 12 | 20 | 0.61 | 2000 | 2000 | 3.0 | 0.7 | 10.8 |
| | 14 | 30 | 0.64 | 3000 | 2500 | 5.0 | 0.8 | 15.2 |
| | 16 | 40 | 0.68 | 5000 | 3000 | 10.0 | 0.9 | 17.9 |
| Thick layer of extra and super heavy oil reservoir | 6.5 | 20 | 0.50 | 10,000 | 2000 | 10.0 | | 7.8 |
| | 8 | 30 | 0.60 | 20,000 | 3000 | 20.0 | | 10.3 |
| | 10 | 40 | 0.70 | 30,000 | 4000 | 30.0 | | 11.0 |
| | 12 | 50 | 0.80 | 40,000 | 5000 | 40.0 | | 11.0 |
| | 14 | 60 | 0.90 | 50,000 | 6000 | 50.0 | | 10.8 |

**Table 8.** Dimensionless data sheet of main control factor analysis.

| Reservoir Types | Reservoir Pressure f | Reservoir Thickness f | Oil Saturation f | Oil Viscosity f | Permeability f | Water Energy f | NTG f | Cumulative Oil Production f |
|---|---|---|---|---|---|---|---|---|
| Single sand body reservoir | 0.59 | 0.21 | 0.85 | 0.28 | 0.12 | 0.03 | | 0.57 |
| | 0.72 | 0.43 | 0.91 | 0.39 | 0.39 | 0.33 | | 0.76 |
| | 0.90 | 0.54 | 0.97 | 0.59 | 0.78 | 0.66 | | 0.96 |
| | 1.08 | 1.07 | 1.03 | 0.79 | 1.18 | 0.99 | | 1.19 |
| | 1.26 | 1.61 | 1.09 | 1.58 | 1.57 | 1.66 | | 1.28 |
| | 1.44 | 2.14 | 1.15 | 2.37 | 1.96 | 2.32 | | 1.24 |
| Layered reservoir | 0.59 | 0.21 | 0.85 | 0.17 | 0.17 | 0.03 | 0.50 | 0.47 |
| | 0.72 | 0.43 | 0.91 | 0.37 | 0.58 | 0.28 | 0.52 | 0.79 |
| | 0.90 | 0.54 | 0.97 | 0.50 | 0.87 | 0.57 | 0.60 | 0.95 |
| | 1.08 | 1.07 | 1.03 | 0.99 | 1.17 | 0.85 | 1.07 | 1.11 |
| | 1.26 | 1.61 | 1.09 | 1.49 | 1.46 | 1.42 | 1.52 | 1.26 |
| | 1.44 | 2.14 | 1.15 | 2.48 | 1.75 | 2.84 | 1.79 | 1.42 |
| Thick layer of extra and super heavy oil reservoir | 0.64 | 0.50 | 0.71 | 0.33 | 0.50 | 0.33 | | 0.76 |
| | 0.79 | 0.75 | 0.86 | 0.67 | 0.75 | 0.67 | | 1.01 |
| | 0.99 | 1.00 | 1.00 | 1.00 | 1.00 | 1.00 | | 1.08 |
| | 1.19 | 1.25 | 1.14 | 1.33 | 1.25 | 1.33 | | 1.08 |
| | 1.39 | 1.50 | 1.29 | 1.67 | 1.50 | 1.67 | | 1.06 |

**Table 9.** Grey correlation calculation results of different types of reservoirs.

| Reservoir Types | Reservoir Pressure | Reservoir Thickness | Oil Saturation | Oil Viscosity | Permeability | Water Energy | NTG |
|---|---|---|---|---|---|---|---|
| Single sand body reservoir | 0.5691 | 0.8491 | 0.7294 | 0.8494 | 0.7572 | 0.7741 | |
| Layered reservoir | 0.6062 | 0.8735 | 0.7732 | 0.8503 | 0.6908 | 0.8203 | 0.8730 |
| Thick layer of extra and super heavy oil reservoir | 0.6832 | 0.8502 | 0.7897 | 0.7232 | 0.7071 | 0.8323 | |

### 3.4. Establishment of Cumulative Oil Production per Well

According to the main control factors selected from the different types of reservoirs, 62 mechanism models were built using the numerical simulation method, forming the chart of the relationship between sensitivity parameters and cumulative oil production per well in different reservoir types and development modes, as shown in Figure 13.

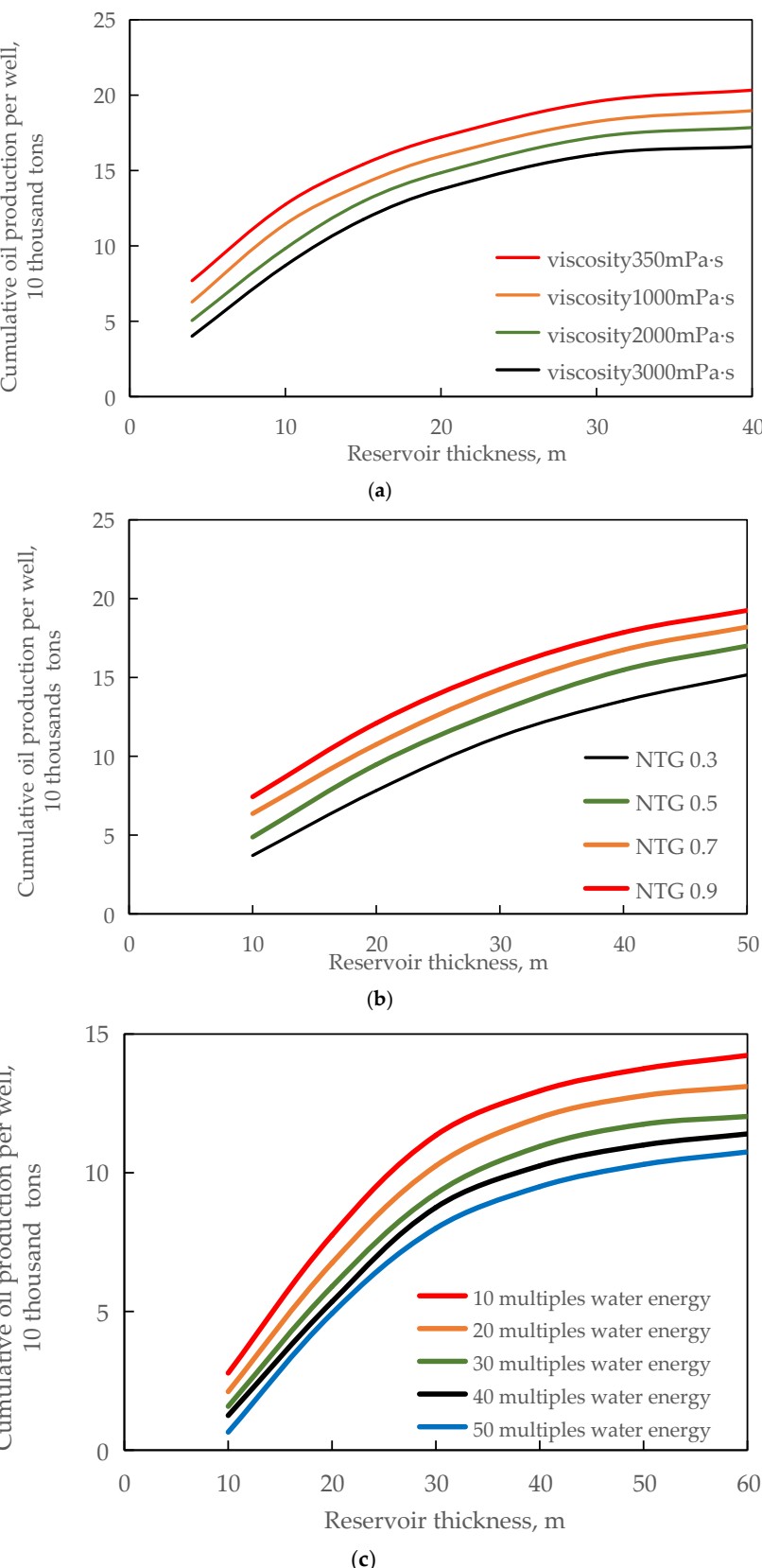

**Figure 13.** Prediction chart of cumulative oil production per well in different types of reservoirs by steam injection. (**a**) Prediction chart of cumulative oil production per well of single sand body reservoirs. (**b**) Prediction chart of cumulative oil production per well of layered reservoirs. (**c**) Prediction chart of cumulative oil production per well of thick layer of extra and super heavy oil reservoirs.

## 4. Economic Evaluation

### 4.1. Method of Economic Evaluation

The economic budget estimate of an offshore heavy oil steam injection project is not only an important basis for determining the project investment and preparing the production plan, but also an important measure for assessing the design scheme. Economic budget estimate is a process in which the reservoir engineers provide developmental indicators, the drilling and completion engineers provide plans and costs, and the offshore platform engineers design different engineering plans according to the development scale and supporting conditions; this information is then submit to the economic engineers to calculate the budget. From the experience of offshore reservoir development, the oilfield scale determines the development investment, and the development investment largely affects the development economy [30,31].

Compared with conventional development, the investment of steam injection is relatively high, while the price of heavy oil is lower than that of light oil, so the minimum economic oil production limit needs to be determined as the basis for deployment. The minimum oil production is calculated according to the input–output method. When the input–output is balanced, that is, the economic benefit is zero, the oil production obtained is the minimum oil production limit:

$$Q_{min} = \frac{C_{fon}}{P_0 R_0 (1 - T_{axo}) - C_{vo}}$$

where $Q_{min}$ is the minimum oil production (10 thousand tons); $C_{fon}$ is the newly increased well drilling and investment to the offshore platform (10 thousand dollars); $P_o$ is the oil price (10 thousand dollars); $R_o$ is the commodity rate of crude oil; $T_{axo}$ is the composite tax rate; and $C_{vo}$ is the operating cost (10 thousand dollars). Among them, crude oil price is calculated at an oil price of 60 USD/barrel based on the evaluation requirement of proved undeveloped oil reserves in China's offshore oilfields. The composite tax rate is 7%. Operating cost refers to the mean value of charges for an offshore heavy oil thermal recovery oilfield that has been put into development.

### 4.2. Basic Mode of Evaluation

According to the scale of the offshore oilfield projects under development, the development investment is mainly affected by the project scale. Combined with the current offshore oilfield development engineering scale, the engineering of offshore heavy oil developed by steam injection can be divided into the following according to reserves and supporting conditions [32,33]: (1) Independent development, building the new offshore wellhead platform and central processing platform, suitable for large-scale development of packaged oilfield above 10 million tons. (2) Relying on development, building the new offshore wellhead platform and mixed transmission manifolds, relying on other oilfields central processing platforms for power and oil processing. It is applicable to the development of oilfields with better supporting conditions and a certain amount of reserves. (3) For further exploitation, only new development wells are drilled and steam injection facilities are added on the original wellhead platform, which is applicable to the oilfields that have been developed and have remaining well slots to improve oil recovery. The project investment and the requirements for reserves is lower. (4) Mobile heat injection, leasing a mobile platform with thermal recovery facilities and building a new offshore wellhead platform with thermal recovery facilities or develop and adjust it within the original platform, which is not only suitable for large-scale development of package reserves, but also suitable for under developed platforms for further exploitation. Engineering facilities under different development modes are designed according to different reserve levels of the oilfield, as shown in Table 10.

**Table 10.** Scales of engineering facilities under different reserve levels and development modes.

| Development Mode | Utilization Reserves (Million Tons) | Standardized Platform | Small Wellhead Platform | Center Processing Platform | Steam Injection Facilities |
|---|---|---|---|---|---|
| Independent development | >20 | Newly build | | | Newly build |
| | 10~20 | | Newly build | Newly build | Newly build |
| Relying on development | >20 | Newly build | | | Newly build |
| | 10~20 | | Newly build | | Newly build |
| Further development | <10 | | | | Newly build |
| Mobile heat injection | | Dependent on reserve scale and supporting conditions | | | |

### 4.3. Application Boundary Study

Based on Formula (3) and defining the reserve levels under different engineering modes, the economic evaluation of the lower limit of cumulative oil production per well with different reserve scales and development modes was carried out, as shown in Figure 14 and Table 10, and the limit of economic cumulative oil production per well of typical reserves is shown in Table 11. According to the scale of the reserves, we can see from the figures and table that the thresholds for independent development and relying on development were higher under different reserve scales. The thresholds of further exploitation and mobile heat injection were lower. When the reserves were less than 10 million tons or more than 20 million tons, the economic oil production limit of mobile heat injection was the lowest. When the reserve level was 10 million tons to 20 million tons, the limit of economic oil production for further development was the lowest. Within the same reserve level, with the increase in producing reserves, the limit of cumulative oil production per well gradually decreased. In addition, the economic production limit of steam flooding after steam huff and puff was lower than that of sidetracking.

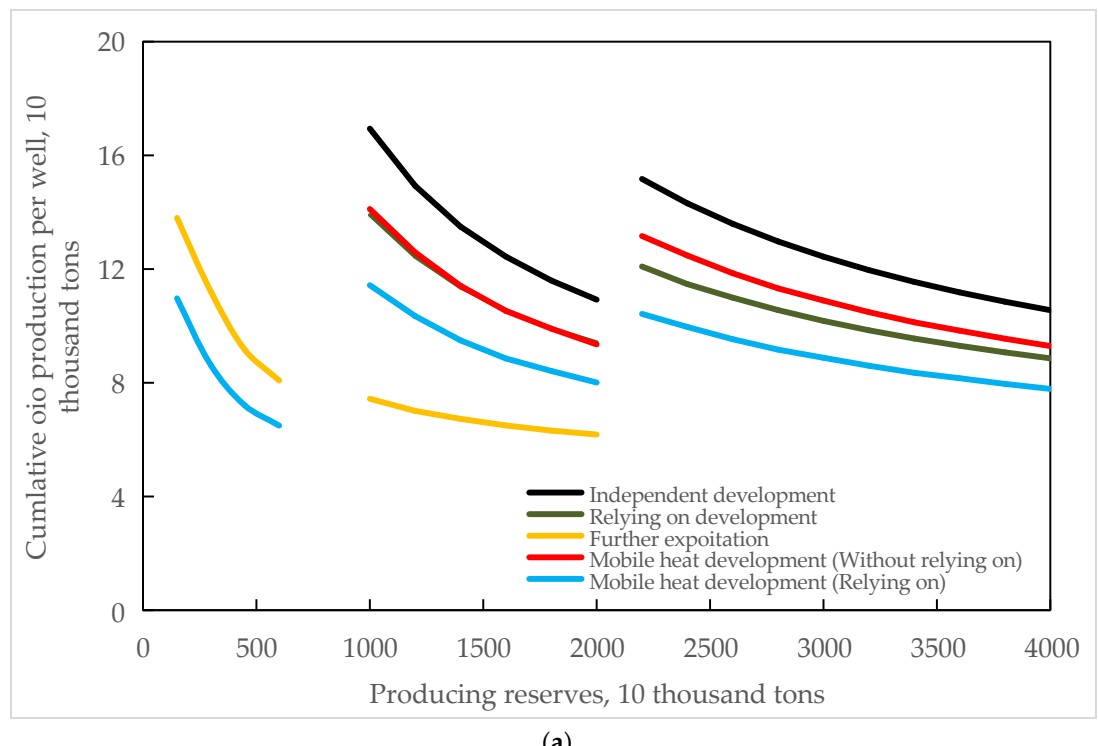

(**a**)

**Figure 14.** *Cont.*

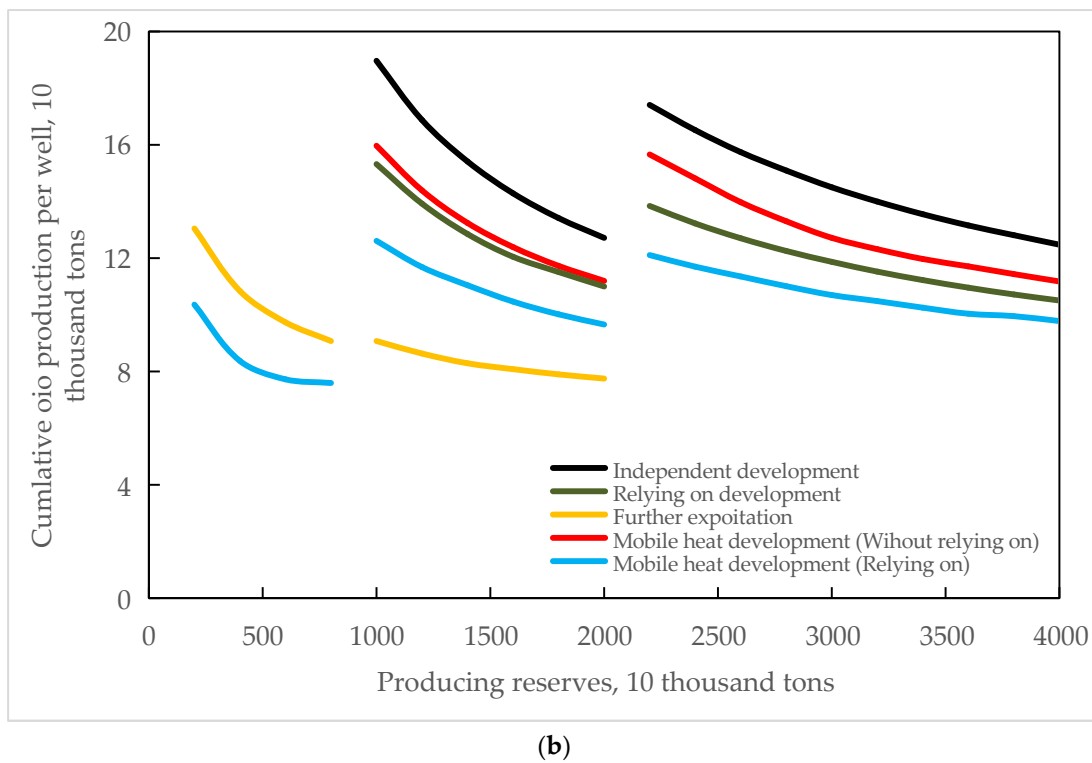

(**b**)

**Figure 14.** Economic oil production limit chart of single well under different reserve scales and different engineering modes. (**a**) Economic oil production limit chart for single well with different reserves and different engineering modes by steam flooding after steam huff and puff. (**b**) Economic oil production limit chart for single well with different reserves and different engineering modes by sidetracking after steam huff and puff.

**Table 11.** The lower limit of economic oil production under different steam injection development modes.

| Producing Reserves, 10 Thousand Tons | Development Modes | Economic Cumulative Production per Well, 10 Thousand Tons | | | | |
|---|---|---|---|---|---|---|
| | | Independent Development | Relying on Development | Further Exploitation | Mobile Heat Injection (Without Relying on) | Mobile Heat Injection (Relying on) |
| 800 | Steam flooding after steam huff and puff | | | 8.1 | | 6.5 |
| 2000 | | 10.9 | 9.4 | 6.2 | 9.4 | 8.0 |
| 4000 | | 10.6 | 8.9 | | 9.3 | 7.8 |
| 1000 | Sidetracking after steam huff and puff | | | 9.7 | | 7.6 |
| 2000 | | 12.7 | 11.2 | 7.8 | 11.0 | 9.7 |
| 4000 | | 12.5 | 10.5 | | 11.2 | 9.8 |

## 5. Discussion and Application

*Development Strategy of Proved Reserves*

According to our research results, the specific reserve development strategies are [34]:

(1)  Combined with the reservoir type and main control factor parameters of proved heavy oil reserves, the cumulative oil production per well can be predicted using the chart of cumulative oil production per well in Figure 12.

(2)  Combined with the development status and offshore development environment around the proved reserves, we can determine the engineering mode that can be adopted in the sea area where the heavy oil reserve is located.

(3)  According to the predicted oil production and the selected engineering mode, the economic production limit under the optimal engineering mode can be inversely deduced by using the economic limit chart in Figure 13. If the economic oil production limit is not higher than the predicted cumulative oil production per well, the

oilfield can realize economic development. On the contrary, it is difficult to realize economic development.

The economic development mode of proved heavy oil reserves is shown in Table 12. According to the above research results, the total amount of undeveloped proved reserves that can be economically developed was 259 million tons, accounting for 44.7% of the total proved reserves. The largest amount of resources that can be developed by mobile heat injection involved 11 oilfields and 97 million tons of proved reserves. At the same time, it can be seen from the table that under the current development strategy and engineering modes, 321 million tons of reserves are still difficult to achieve economic exploitation.

**Table 12.** Engineering model planning for economic production of proved heavy oil reserves in Bohai Bay.

| Development Mode | Producing Reserves Million Tons | Number of Oilfield Involved | Oilfield Involved |
| --- | --- | --- | --- |
| Independent development | 82 | 2 | JZ23-2, LD5-2N |
| Relying on development | 25 | 1 | KL9-6 |
| Further exploitation | 55 | 4 | LD27-2, LD16-3, etc. |
| Mobile heat injection | 97 | 11 | PL19-3, QHD33-1S, etc. |
| Difficult to realize economic development at present | 383 | | |

According to the production profile prediction of economically available reserves, the peak capacity of steam injection development of Bohai heavy oil can contribute to 2.78 million tons, and the cumulative oil production in 23 years is predicted to be 32 million tons. As shown in Figure 15.

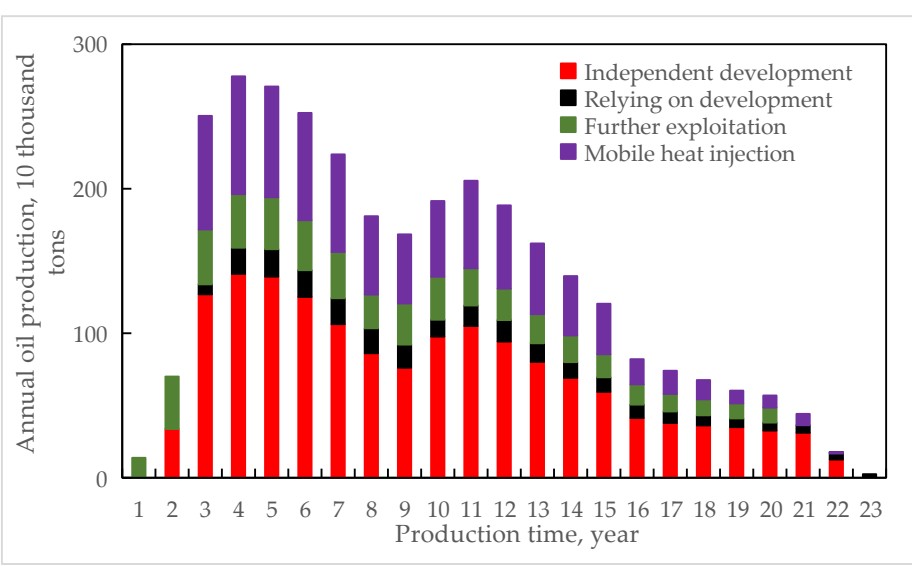

**Figure 15.** Production planning for steam injection of undeveloped heavy oil reserves in the Bohai oilfield.

From the perspective of economic development planning of heavy oil reserves, it is still difficult to achieve economic development under the current development modes for more than half of the reserves. In order to improve the production rate of heavy oil reserves and speed up the production capacity construction, the key directions for improving quality and efficiency are accelerating the pace of testing new technologies; exploring the technology of continuous enhanced oil recovery after steam injection development, for example, in situ combustion, SAGD, supercritical technology, etc.; exploring low-cost thermal recovery methods, such as underground thermal generation and hot water injection; exploration of

drilling and completion and engineering cost reduction modes, such as longer horizontal well technology; increasing the speed and efficiency-of technology for drilling and completion; revamping of old drilling offshore ship; miniaturization of platform facilities; and development of higher efficiency processes to address key problems [35–38].

## 6. Conclusions

(1) Based on the laboratory physical simulation experimental method, the potential of superheated steam development in offshore reservoirs was identified. The numerical simulation equations for heavy oil steam injection development were established, and the matching error of the experimental results was under 10%.

(2) From the numerical simulation comparison, superheated steam flooding after superheated steam huff and puff in single sand body reservoir and layered reservoir, and sidetrack after superheated steam huff and puff in extra and super heavy oil reservoirs were identified as the optimum development modes. The main factors influencing cumulative oil production of steam injection development in different reservoir types were screened by the numerical simulation method and grey correlation method, and prediction charts of cumulative oil production per well were established.

(3) According to the discussion of reserve classification, the economic oil production limit charts for a single well of the different engineering models by the offshore economic evaluation method were established. Compared with other engineering modes, further exploitation and mobile heat injection were lower. At the end of the paper, the economic development mode of proved heavy oil reserves was planned. A total of 18 oilfields or blocks can achieve economic development in different modes, with a cumulative developed reserves of 259 million tons and a peak capacity of 2.78 million tons, which provides a decision for the construction of steam injection capacity of the Bohai heavy oil fields.

(4) Under the current development strategy and engineering modes of offshore heavy oil, it is still difficult to achieve economic development for more than half of the heavy oil reserves. In order to reduce the threshold of economic development, offshore heavy oil should improve quality and efficiency in terms of development mode, cost reduction of drilling and production engineering, and optimization of thermal recovery processes.

**Author Contributions:** Methodology, T.W.; Software, F.L.; Writing—review & editing, X.L. All authors have read and agreed to the published version of the manuscript.

**Funding:** The research was funded by the major project of Science and Technology of China National Offshore Oil Corporation (YXKY—ZX 06 2021).

**Data Availability Statement:** Not applicable.

**Conflicts of Interest:** The authors declare no conflict of interest.

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
