# Peer review of "Optimization of Efficient Development Modes of Offshore Heavy Oil and Development Planning of Potential Reserves in China"

_water, doi:10.3390/w15101897_

Round 1

Reviewer 1 Report

Scientific work is devoted to increasing the efficiency of development, production of heavy oil. The article has a pronounced applied orientation. It has a sufficient amount of graphic material and supporting results. However, the article has a number of serious remarks.

1. There is no clear statement of the task in the administration. There is no reasoning for the importance of the study. A small attempt was made to review existing studies. The author is encouraged to revise the introduction, indicating a clear need and importance of the study.

2. The article completely lacks a methodological review. The authors do not analyze alternative methods. Do not render the results of other studies. The presented work is similar to a report on the results of laboratory studies. After the introduction, the author needs to add a section devoted to the analysis of the subject area. Show alternative and existing methods of extraction, evaluation, exploitation of oil fields. It would be useful for the authors to read this study https://doi.org/10.1038/s41598-022-21778-0. including it in your review

3. In offshore development conditions, it is much more efficient to use inhibitors. In the experiment, you only describe steam. But pumping steam to great depths is not effective, how do you justify this?

4. 118-133 What are the errors of your laboratory setup? If they are scaled down, how? What is the ratio of the real object and your laboratory setup?

5. Table 2-4 explain "saturated hot water (250℃)" "saturated steam (250℃)" what is the difference in these terms for your lab setup?

6. There is no discussion section in the article. The author needs to add a discussion section before the conclusion. In which it is necessary to discuss the results obtained, highlighting the positive and negative aspects of the study. Indicating directions for further research or indicating their completion.

7. Scientific research, as a rule, is accompanied by a list of references. The author needs to expand the list of references to at least 35-40 sources by adding missing sections.

After eliminating these shortcomings, the article can be accepted for publication.

Author Response

  1. There is no clear statement of the task in the administration. There is no reasoning for the importance of the study. A small attempt was made to review existing studies. The author is encouraged to revise the introduction, indicating a clear need and importance of the study.

Response: Thanks for your suggestion and will carefully revise the introduction.

  1. The article completely lacks a methodological review. The authors do not analyze alternative methods. Do not render the results of other studies. The presented work is similar to a report on the results of laboratory studies. After the introduction, the author needs to add a section devoted to the analysis of the subject area. Show alternative and existing methods of extraction, evaluation, exploitation of oil fields. It would be useful for the authors to read this study https://doi.org/10.1038/s41598-022-21778-0. including it in your review

Response: Thanks for your suggestion. I will carefully read this article and make modifications

  1. In offshore development conditions, it is much more efficient to use inhibitors. In the experiment, you only describe steam. But pumping steam to great depths is not effective, how do you justify this?

Response: Thanks for your suggestion. From the current development progress of offshore heavy oil thermal recovery, the bottomhole steam quality can generally reach above 0.8, and the supporting drilling and completion technology is already very complete.

  1. 118-133 What are the errors of your laboratory setup? If they are scaled down, how? What is the ratio of the real object and your laboratory setup?

Response: Thanks for your suggestion. The example design of similar proportions is shown in the table below.

Parameter

reservoir type

lab type

Basic parameter

reservoir thickness(m,cm)

10

5

porosity(%)

35

35

permeability(mD)

4980

40000

initial oil saturation(%)

80

80

Irreducible water saturation(%)

20

20

reservoir temperature(℃)

50

50

oil viscosity(mPa·s)

627

627

injection temperature(℃)

280

280

steam quality

0.8

0.8

rock Thermal diffusion coefficient (m2/h)

0.00197

0.00197

Steam huff and puff

cyclic steam injection volum(m3,mL/min)

3000

150

soak time(d,min)

3

2

production date(d,min)

100~

20

Steam flooding

steam injection rate(m3/d,mL/min)

250-300

30-40

injection time(a,min)

1

59.5

  1. Table 2-4 explain "saturated hot water (250℃)" "saturated steam (250℃)" what is the difference in these terms for your lab setup?

Response: “saturated hot water” is the water in a critical state with the quality of steam is 0, while “saturated steam” is the steam with the quality of steam is 1. the steam has more latent heat of vaporization than water.

  1. There is no discussion section in the article. The author needs to add a discussion section before the conclusion. In which it is necessary to discuss the results obtained, highlighting the positive and negative aspects of the study. Indicating directions for further research or indicating their completion.

Response: Thanks for your suggestion, we will consider adding some discussion results.

  1. Scientific research, as a rule, is accompanied by a list of references. The author needs to expand the list of references to at least 35-40 sources by adding missing sections.

Response: Thanks for your suggestion, we will add some sources.

After eliminating these shortcomings, the article can be accepted for publication.

Reviewer 2 Report

(1) The geological overview and development status of the study area need to be introduced in the manuscript. In this way, readers can refer to the relevant data in this study if they want to carry out numerical simulation of heavy oil development in the study area.

(2) It can be seen from reading the manuscript that the rock samples used in the study were prepared manually. However, how to ensure that the prepared rock samples can restore the characteristics of the actual rock samples? Just based on porosity and permeability characteristics? Here, I suggest adding the grain size and composition of the actual rock samples taken from the oil field. In this case, it is relatively persuasive and pertinent. Also, the experimental parameters described in the experimental procedure (section 2.3) are very vague, please be specific (or give a parameter table).

(3) Although the model used for the numerical simulation is described in Section 3.1.1, the details of the model (such as definite solution conditions, etc.) have not been given a detailed description. Please add this point so that the reader can restore the simulation process efficiently.

(4) What is the purpose and basis of designing the injection medium to include normal temperature water (25 ℃), hot water (120 ℃), saturated hot water (250 ℃), steam (250 ℃) and superheated steam (300 ℃)? Why not choose another temperature? Moreover, why three types of heavey oil were chosen herein? I think one oil is OK!

(5) As we all know, the fluid flow in porous media is extremely complex, while the curve in Fig. 12a is almost parallel! Reasonably, this is not quite in line with common sense. Why does this happen?

(6) There are also some small nibs needs to be modified. For example, In Figure 13, the blue curve in the legend disappears. In Fig 10, Fig 6, Fig 5, Fig 4, the coordinate axis and legend need to be adjusted to the proper position! what is the title for the last figure in manuscript?

(7) It was mentoned that "Due to the increasing domestic demand for energy and the increasing difficulty of heavy oil exploitation in the new exploration of onshore oilfields in China, how to efficiently utilize offshore heavy oil reserves has gradually attracted the attention of scholars all over the world." in manuscript. Some references need to be cited for surport it, as follows: â‘ Environmental Science and Pollution Research, 2022, 29(51): 77737-77754; â‘¡ Journal of Molecular Liquids, 376: 121394; â‘¢Applied Energy, 239: 1190-1211.

Author Response

  • The geological overview and development status of the study area need to be introduced in the manuscript. In this way, readers can refer to the relevant data in this study if they want to carry out numerical simulation of heavy oil development in the study area.

Response: Thanks for your your suggestion, we will add the introduce of offshore heavy oil development progress.

  • It can be seen from reading the manuscript that the rock samples used in the study were prepared manually. However, how to ensure that the prepared rock samples can restore the characteristics of the actual rock samples? Just based on porosity and permeability characteristics? Here, I suggest adding the grain size and composition of the actual rock samples taken from the oil field. In this case, it is relatively persuasive and pertinent. Also, the experimental parameters described in the experimental procedure (section 2.3) are very vague, please be specific (or give a parameter table).

Response: Thanks for your suggestion, the porosity and permeability of rocks in Bohai Oilfield have a strong correlation when characterizing the pore structure characteristics. Generally, we use a few certain meshes of sand, such as 80 mesh, 120 mesh, etc., to mix in proportion.

  • Although the model used for the numerical simulation is described in Section 3.1.1, the details of the model (such as definite solution conditions, etc.) have not been given a detailed description. Please add this point so that the reader can restore the simulation process efficiently.

Response: Thanks for your suggestion, we will consider adding a detailed process in the future.

  • What is the purpose and basis of designing the injection medium to include normal temperature water (25 ℃), hot water (120 ℃), saturated hot water (250 ℃), steam (250 ℃) and superheated steam (300 ℃)? Why not choose another temperature? Moreover, why three types of heavey oil were chosen herein? I think one oil is OK!

Response: Thanks for your suggestion, 25 ℃ is the temperature of normal oil for water injection, 120 ℃ is the temperature at the bottom of the hot medium boiler injection well, 250 ℃ is the saturation temperature of the reservoir at low pressure during the steam flooding, 300 ℃ is the initial injection temperature of CSS. Three types of oil experiments were conducted to explore reasonable development methods for heavy oil with different crude oil viscosities.

  • As we all know, the fluid flow in porous media is extremely complex, while the curve in Fig. 12a is almost parallel! Reasonably, this is not quite in line with common sense. Why does this happen?

Response: Thanks for your suggestion. There are two reasons. The first is the mechanism model were used to study the quantitative description of the bottom hole. The second is that the relationship between bottom hole temperature and heterogeneity is not significant. The figure is similar to the measured bottom hole temperature of offshore developed heavy oil reservoirs.

  • There are also some small nibs needs to be modified. For example, In Figure 13, the blue curve in the legend disappears. In Fig 10, Fig 6, Fig 5, Fig 4, the coordinate axis and legend need to be adjusted to the proper position! what is the title for the last figure in manuscript?

Response: Thanks for your suggestion. The last figure is the future production planning of Bohai Oilfield.

  • It was mentoned that "Due to the increasing domestic demand for energy and the increasing difficulty of heavy oil exploitation in the new exploration of onshore oilfields in China, how to efficiently utilize offshore heavy oil reserves has gradually attracted the attention of scholars all over the world." in manuscript. Some references need to be cited for surport it, as follows: â‘ Environmental Science and Pollution Research, 2022, 29(51): 77737-77754; â‘¡ Journal of Molecular Liquids, 376: 121394; â‘¢Applied Energy, 239: 1190-1211.

Response: Thanks for your suggestion. We will consider adding these high quality papers.

Reviewer 3 Report

The work is devoted to the development of heavy oil fields. The scientific and practical significance of this work lies in the use of the mathematical and methodological apparatus of system analysis. The work is of wide scientific interest, but has a number of significant shortcomings.

1. One of the main requirements of the journal is the presence of a strict structure for the presentation of material. The paper does not contain a section "description of the subject area" and a section "discussion".

2. Figure 11 shows the regression model. It must be included in the text of the work, removed from the figure. For other figures where regression models are obtained, they must also be removed from the figures in the text of the work.

3. The list of references does not correspond to the scientific publication. The author needs to expand the list to 40 sources. The author is recommended to do this by highlighting the section "description of the subject area"

4. It would be useful to expand the scientific part of the work by analyzing oil with a high ARPD content. The journal https://www.mdpi.com/1996-1073/15/17/6462 describes this direction very well. The author is encouraged to view this work. Referring to it expand your literary review. Also get acquainted with the works of such authors as: Kukharova, T.V.; Martirosyan, A.V.; Nerman, H.; Fathi, M.; Soliman, S.; El Maghraby, H.; Mustfa, Y.M. Ragunathan, T.; Husin, H.; Wood, C.D. and others. Their work can be found on the Internet or in the references of the work indicated above.

After these shortcomings are eliminated, the work can be accepted for publication.

Author Response

  1. One of the main requirements of the journal is the presence of a strict structure for the presentation of material. The paper does not contain a section "description of the subject area" and a section "discussion".

Response: Thanks for your suggestion, We will consider increasing the length of this section later.

  1. Figure 11 shows the regression model. It must be included in the text of the work, removed from the figure. For other figures where regression models are obtained, they must also be removed from the figures in the text of the work.

Response: Thanks for your suggestion, we will delete the regression formula.

  1. The list of references does not correspond to the scientific publication. The author needs to expand the list to 40 sources. The author is recommended to do this by highlighting the section "description of the subject area"

Response: Thanks for your suggestion, we will consider adding some content.

  1. It would be useful to expand the scientific part of the work by analyzing oil with a high ARPD content. The journal https://www.mdpi.com/1996-1073/15/17/6462 describes this direction very well. The author is encouraged to view this work. Referring to it expand your literary review. Also get acquainted with the works of such authors as: Kukharova, T.V.; Martirosyan, A.V.; Nerman, H.; Fathi, M.; Soliman, S.; El Maghraby, H.; Mustfa, Y.M. Ragunathan, T.; Husin, H.; Wood, C.D. and others. Their work can be found on the Internet or in the references of the work indicated above.

Response: Thanks for your suggestion, we will fully learn and absorb the views of these articles.

Round 2

Reviewer 1 Report

All questions have been fully answered by the author. Notice no. The article can be recommended for publication in its present form.

Author Response

Thanks for your approval of the artical!

Reviewer 2 Report

(1) There are some grammatical inadequacies in the manuscript, and the language of the manuscript needs further polishing by native English speakers or professional institutions.

(2) The reference Journal of Molecular Liquids, 376: 121394. also needs to be cited in manuscript. 

Author Response

Thanks for your support and approval! The modification method of some statements has been modified, and the reference had been added to the artical.